# A Rare Case of Bickerstaff Encephalitis in Childhood: A Case Report

**DOI:** 10.3390/neurolint17020024

**Published:** 2025-02-07

**Authors:** Luca Gregorio Giaccari, Donatella Mastria, Rosella Barbieri, Rossella De Maglio, Francesca Madaro, Gianfranco Paiano, Luciana Mascia, Maria Caterina Pace, Giuseppe Pulito, Pasquale Sansone

**Affiliations:** 1Department of Anesthesia and Intensive Care, “Vito Fazzi” Hospital, 73100 Lecce, Italy; lucagregorio.giaccari@gmail.com (L.G.G.); donatella.mastria71@libero.it (D.M.); rosella.barbieri@hotmail.it (R.B.); rossella.demaglio@gmail.com (R.D.M.); madaro.francesca1@gmail.com (F.M.); gianfrancopaiano87@gmail.com (G.P.); pulitogiuseppe00@gmail.com (G.P.); 2Department of Woman, Child and General and Specialized Surgery, University of Campania “Luigi Vanvitelli”, 80138 Naples, Italy; mariacaterina.pace@unicampania.it; 3Department of Experimental Medicine, University of Salento, 73100 Lecce, Italy; luciana.mascia@unisalento.it

**Keywords:** bickerstaff brainstem encephalitis, anti-GQ1b antibody, anti-GM antibody, childhood

## Abstract

**Background:** Bickerstaff brainstem encephalitis (BBE) is a rare autoimmune disease and approximately 74 cases have been reported in the literature, mostly in childhood. **Methods:** We reported this case report according to the CARE guidelines. **Results:** A 13-year-old female presented with a 4-day history of persistent fever and hallucinations. She rapidly developed nystagmus associated with blurred vision with ataxic gait. She also developed altered mental status, blepharoptosis, diplopia and extrinsic ocular motility. An EEG showed asymmetric brain electrical activity with slow and spiky abnormalities in the left cerebral hemisphere. Lumbar puncture showed mild pleocytosis with lymphocytic predominance, elevated protein, with normal glucose. Anti-GM1 and anti-GM2 antibodies were positive. She was administered intravenous immunoglobulin therapy due to a suspicion of BBE, showing rapid improvement in mental status. **Conclusions:** BBE is a diagnosis of exclusion and should be considered especially in pediatric age.

## 1. Introduction

The first case of Bickerstaff brainstem encephalitis (BBE) was reported by Bickerstaff and Cloake in 1951 [1]. Later, in 1957, Bickerstaff reported the first three cases in childhood and renamed the condition “brainstem encephalitis” [2].

BBE is a type of brainstem encephalitis with specific neurological deficits and its incidence rate during childhood is unknown [3,4,5]. Among adults, a Japanese national survey estimated the annual incidence to be approximately 0.078 per 100,000 population [6]. Due to its rarity, it is difficult to clarify the correct etiology (direct infection, autoimmune demyelination or post-infectious reaction, or vasculitis) and, therefore, the most appropriate diagnostic framework and therapeutic treatment.

To the best of our knowledge, only a few cases of BBE in childhood are reported in the literature [7]. The aim of this work is to show a case of BBE in childhood that is negative for serum IgG anti-GQ1b antibodies.

We reported this case report according to the CARE guidelines [8,9]. The ethics committee was not required, in accordance with the most recent Italian guidelines, which establish that approval by the ethics committee is not required for a maximum sample of three to five patients and where there are no deviations from common clinical practice.

## 2. Case Report

A 13-year-old female without a previous history of disease presented to the emergency department (ED) of our hospital with a 4-day history of persistent fever and hallucinations. She reported also having abdominal pain the days before. No history of respiratory or gastrointestinal tract infection was reported.

On arrival to ED, the level of consciousness was normal. Her vital signs were as follows: heart rate (HR) 105 beats per minute, blood pressure (BP) 145/83 mmHg, oxygen saturation (SaO2) 99% and body temperature (BT) 39.8 °C. Physical examination revealed pain on palpation in the right iliac fossa. Blood tests revealed elevated levels of white blood cells (WBC, 12.870 μL) and procalcitonin (PCT, 0.9 ng/mL). Findings on chest radiography and abdominal ultrasound were unremarkable. A nasopharyngeal swab was performed with a positive result for adenovirus on real-time reverse transcriptase–polymerase chain reaction assay.

The patient was admitted to the pediatric ward with a diagnosis of suspected upper respiratory infection and empiric antimicrobial therapy was started.

During the first days of her hospitalization, the patient presented with a fever. Nystagmus associated with blurred vision and ataxic gait were observed on day 4. Cerebral computed tomography (CT) was normal; no brain abnormalities were detected on brain MRI in addition to previous examinations. An abnormal electroencephalogram (EEG) was registered showing asymmetric cerebral electrical activity with slow and pointed anomalies in the left cerebral hemisphere.

Lumbar puncture (LP) was performed on day 5 and cerebrospinal fluid (CSF) analysis showed a slight pleocytosis of 42 cells/uL (reference 0–10 cells/uL) with lymphocytic predominance, elevated protein of 51 mg/dL (reference 15–45 mg/dL), and normal glucose of 0.73 g/L (reference 0.40–0.70 g/L) (see Table 1). CSF PCR was negative for the bacteria and viruses tested. Brain magnetic resonance imaging (MRI) did not show any abnormalities. Thyroid function was normal. Blood serological tests for hepatitis A (HAV), B (HBV) and C (HCV), Human Immunodeficiency Virus (HIV), and syphilis were negative. Treponema pallidum came negative, as well as Varicella-zoster virus (VZV), Epstein Barr virus (EBV), Cytomegalovirus (CMV), Mycoplasma pneumoniae, and Borrelia burgdorferi.

The patient was started on IV ceftriaxone and IV dexamethasone given the suspicion of meningoencephalitis without notable improvement. For the appearance of right clonic seizures, an EEG was repeated on day 6 showing slowed and asymmetric rhythm due to the appearance of theta delta slow potentials with a pointed morphology and subcontinuous character on the left fronto-central-temporal sites with a tendency to contralateral diffusion. Seizures were treated with IV midazolam boluses and treatment with IV levetiracetam 250 mg twice a day was started.

Due to the persistence of seizures despite medical therapy, the patient was admitted to our Intensive Care Unit (ICU). On admission, she presented alteration of mental state (GCS 14/15 [E3 V5 M6]), blepharoptosis, nystagmus, diplopia, and anomalies of extrinsic ocular motility. Continuous infusion of midazolam was administered and for persistence of focal clonic seizures on day 9, perampanel 2 mg was started for 5 days and subsequently 4 mg. Serum and CSF autoimmune encephalitis panel was performed.

On day 11, a cerebral MRI was repeated showing predominantly left frontal cerebritis foci (see Figure 1).

On day 13, hearing loss occurred. Otoacoustic emissions (OAE) and auditory evoked potentials (AEPs) were performed showing absence of response bilaterally. On day 16, somatosensory evoked potentials (SEPs) were performed showing for the left upper and lower limb “cortical latency responses poorly represented on the left” and for the right upper and lower limb “cortical responses of slightly increased latency on the right”.

Serum anti-GM1 and anti-GM2 antibody came positive. Detection of oligoclonal IgG bands (OCBs) in parallel cerebrospinal fluid (CSF) and serum samples supported our diagnosis of neuroinflammatory disease.

BBE was suspected, for which the patient was started on intravenous immunoglobulin (IVIG) 0.4 g/kg/die on day 14. After 5 days, the patient’s mental status improved and she was discharged to the pediatric ward on hospital day 25.

The patient was discharged from hospital after 45 days. She achieved complete recovery except for severe bilateral hearing loss.

## 3. Discussion

Our case reports a rare case of BBE in childhood. As described in a previous review, 74 cases are reported in the literature [7].

BBE is an autoimmune disease that is part of the same spectrum as Miller–Fisher syndrome and Guillain–Barré syndrome.

As for other neurological syndromes, the etiology is unfortunately unknown [10]. In 50% of cases, BBE occurs following respiratory or gastrointestinal tract infections, most frequently caused by M. pneumoniae, H. influenzae, and C. jejuni [11,12]. In our case, the results of a nasopharyngeal swab indicated adenovirus infection. Adenoviruses commonly cause respiratory infections, conjunctivitis, pharyngitis, and gastroenteritis, particularly in children. In immunocompetent children, adenovirus is a rare cause of central nervous system (CNS) disease. Manifestations can range from mild forms, such as aseptic meningitis, to severe and potentially fatal forms, such as acute necrotizing encephalopathy [13].

The child in our clinical case had typical symptoms of BBE. Clinical symptoms of BBE mainly include consciousness disturbance, gait disturbance, and external ophthalmoplegia [14]. Ocular motor weakness and ptosis, facial weakness, and bulbar weakness are common manifestations. Extensor plantar response (Babinski’s sign) could be present. Deep tendon reflexes are usually absent or decreased, but can be normal or brisk. Sensory disturbance and autonomic instability may occur [15].

There is no gold standard for the diagnosis of BBE. Lumbar puncture is a high priority test for suspected central nervous system disease [16]. CSF analysis shows albuminocytological dissociation or pleocytosis [15,17]. Abnormal EEG findings indicated central nervous system involvement, demonstrating impaired consciousness. EEG changes correlate with the level of consciousness in patients with BBE and often show predominant N1 and/or N2 sleep patterns [18]. As shown previously, these characteristic EEG changes are possibly caused by dysfunction of the ascending reticular activating system (ARAS) [18]. ARAS is part of the brainstem reticular formation, and dysfunction of which is considered to be the cause of impaired consciousness in BBE. SEPs results suggest the interruption of the somatosensory pathway within the brainstem. MRI is the gold standard technique for brain imaging in encephalitis [19]. In BBE patients abnormal MRI lesions, such as high-intensity areas on T2-weighted images of the brainstem, thalamus, cerebellum and cerebrum, are characteristics [20]. All of these diagnostic data were found in our case report.

Anti-GQ1b antibodies are detected in more than half of the patients [18]. The presence of anti-GM1 antibodies is detected in almost 40% of patients. The role of anti-ganglioside antibodies in the pathogenesis of BBE is not completely understood. Gangliosides are sphingolipid compounds that constitute cell membranes, particularly neurons. They are abundant in synapses, where they play a fundamental role in communication between nerve cells. There are more than 20 different gangliosides, and autoantibodies against these molecules are found in numerous diseases of the nervous system (e.g., inflammatory neuropathies, Miller–Fisher syndrome, Guillain–Barré syndrome). The anti-ganglioside antibodies could participate in direct damage to the structure that they bind to, be a consequence of several types of infectious diseases, or facilitate many immune-mediated pathological mechanisms [12]. Our patient was negative for anti-GQ1b antibodies; however, interestingly, she was positive for anti-GM1 and anti-GM2 antibodies.

Furthermore, oligoclonal bands in the CSF are an indicator of intrathecal synthesis of IgG and IgM immunoglobulins in the central nervous system [21]. As in our case, the presence of oligoclonal bands in the CSF supports the hypothesis that an autoimmune mechanism underlies BBE pathogenesis. So in doubtful cases, CSF oligoclonal banding of the cerebrospinal fluid could be a useful test to look for proteins present during the inflammation process of the CSF.

In our case, the diagnosis of BBE was based on typical symptoms, such as external ophthalmoplegia, ataxia, and disturbance of consciousness. Typical MRI images (high signal lesion on T2-weighted images located in the brainstem) and high titers of anti-ganglioside antibodies may contribute to the diagnosis of BBE.

For a definitive diagnosis, it is necessary to exclude other conditions, such as Wernicke’s encephalopathy, cerebrovascular disorder, multiple sclerosis, neuromyelitis optica, myasthenia gravis, brainstem tumor, vasculitis, botulism, and Hashimoto encephalopathy.

There is certainly overlap between Guillain–Barré syndrome, Miller–Fisher syndrome, and BBE, as well as other conditions associated with anti-ganglioside antibodies, such as chronic ophthalmoplegia with anti-GQ1b antibodies and the pharyngo-cervico-brachial variant of GBS [22]. Indeed, they are all post-infectious disorders that share common clinical features, such as ataxia and ophthalmoplegia. There is often a prodromal upper respiratory tract infection, CSF albumin-cytologic dissociation, and serum anti-ganglioside antibodies. However, BBE is a central nervous system disease, while Guillain–Barré syndrome and Miller–Fisher syndrome are peripheral nervous system disorders.

There are no established guidelines for treatment [20]. Autoimmune encephalitis, including BBE, responds to immunomodulatory therapy, such as IVIG and plasmapheresis [23]. This treatment is accompanied by the administration of intravenous corticosteroids [23]. IVIG are typically administered at high doses and provide various anti-inflammatory and immunomodulatory effects. The administration of 2 g/kg for 5 days (0.4 g/kg/day) is planned. IVIG can be used as monotherapy or in combination with plasmapheresis. Plasmapheresis effectively removes autoantibodies and other pathological substances in plasma. A session is planned every other day for 5–7 cycles. Among corticosteroids, methylprednisolone is the most commonly used at a dose of 1 g daily, for 3–5 days. For our patient, IVIG administration was chosen and the patient’s symptoms gradually improved after therapy; plasma exchanges were considered as secondary treatment due to the young age of the patient and the risks associated with the use of a large bore catheter. Plasmapheresis is an invasive procedure with common adverse effects, occurring up to twice as often in children than in adults [24]. Complications of central venous access (e.g., thrombosis, accidental line displacement, and line infection), depletion of non-pathogenic blood components (e.g., coagulation factors, especially fibrinogen), electrolyte imbalances (e.g., hypocalcemia, hypomagnesemia, hypokalemia, and metabolic alkalosis), and hypotension are quite common. However, all of these possible complications are usually mild and resolvable.

BBE has a good prognosis and its course is generally monophasic. The recovery in childhood is faster than in adulthood, usually resolving within 4 to 6 weeks after starting treatment [3,5]. Most patients recover completely within 6 months. In our case, recovery was complete except for mild hearing loss. The altered auditory brainstem response probably reflects activation from the cochlea to the midbrain and may be caused by small lesions located especially in the pons [11].

Controlled studies to demonstrate the diagnostic pathway and efficacy of various treatments (IVIG versus plasmapheresis) in BBE would likely be logistically impractical, due to the low incidence of the disorder in childhood. When encountering a rare pathology, such as BBE, case reports are valuable sources of unusual information that can lead to advances in daily clinical practice.

## 4. Conclusions

In conclusion, we reported a case of BBE in childhood, which needs to be brought to the attention of clinicians. Bickerstaff encephalitis is a diagnosis of exclusion that should be considered in the presence of acute brainstem dysfunction, especially after an infectious syndrome without other obvious etiologies, especially in pediatric age.

## Figures and Tables

**Figure 1 neurolint-17-00024-f001:**
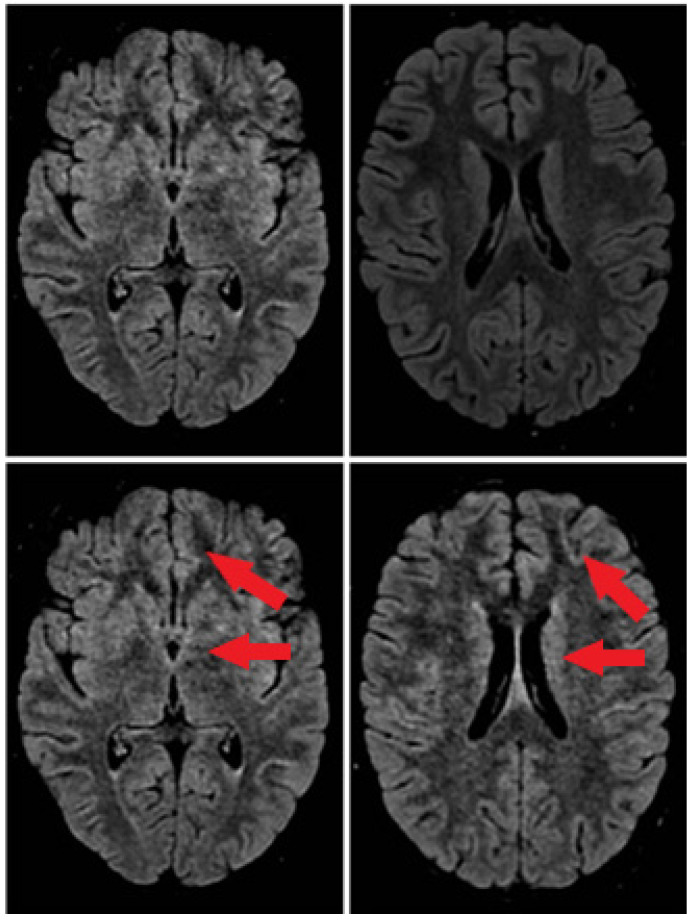
Brain MRI. At the onset of the disease (first line). After 11 days of disease (second line).

**Table 1 neurolint-17-00024-t001:** CSF analysis.

		Normal Range
Color	clear	clear
CSF glucose	0.73	0.40–0.70 g/L
CSF proteins	51	15–45 mg/dL
CSF albumin	37.8	13.9–24.6 mg/dL
WBCs count	42	0–10 elements/μL
RBCs count	Nil	0–10 elements/μL
Microbial examination	No microorganism	

CSF, cerebrospinal fluid; WBCs, white blood cells; RBCs, red blood cells.

## Data Availability

The original contributions presented in the study are included in the article. Further inquiries can be directed to the corresponding author.

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
