# Peer review of "A Rare Case of Bickerstaff Encephalitis in Childhood: A Case Report"

_2035-8377, 2025, doi:10.3390/neurolint17020024_

Round 1
Reviewer 1 Report
Comments and Suggestions for Authors
This is a well written case report. Perhaps it is the small size of the image, but I am not able to appreciate cerebritis on the MRI. Please provide a larger image with arrows pointing to abnormalities.
- The main issue discussed is a case of Bickerstaff encephalitis in a 13 year old patient.
- Since pediatric cases of this condition are rare, the topic is relevant to the field of neuroimmunology.
- The paper adds a well-described case to a limited literature.
- The data substantiates the diagnosis and the narrative reviews the literature.
- The references are appropriate.
- The image provided is not satisfactory. It should be larger and should have arrows pointing to evidence of cerebritis.
Author Response
Dear reviewer,
the image has been modified.
Reviewer 2 Report
Comments and Suggestions for Authors
Bickerstaff encephalitis in childhood: a review of 74 cases in the literature from 1951 to today. Luca Gregorio Giaccari1* Donatella Mastria1 Rosella Barbieri1 Rossella De Maglio1 et al

Extensive editing needed
Author Response
Dear reviewer,
- the abstract has been modified as requested;
- the introduction has been modified as requested;
- "case presentation" has been modified to "case report";
- the discussion has been modified as requested;
- the conclusion has been modified as requested:
- ref 3 is appropriate regarding cases of BBE in childhood described in the literature.